# Post-Diagnosis Dietary Patterns among Cancer Survivors in Relation to All-Cause Mortality and Cancer-Specific Mortality: A Systematic Review and Meta-Analysis of Cohort Studies

**DOI:** 10.3390/nu15173860

**Published:** 2023-09-04

**Authors:** Maria-Eleni Spei, Ioannis Bellos, Evangelia Samoli, Vassiliki Benetou

**Affiliations:** Department of Hygiene, Epidemiology and Medical Statistics, School of Medicine, National and Kapodistrian University of Athens, 75 Mikras Asias Street, 115 27 Athens, Greece; marilena_0108@hotmail.com (M.-E.S.); bellosg@windowslive.com (I.B.); esamoli@med.uoa.gr (E.S.)

**Keywords:** dietary patterns, cancer survivors, a priori dietary patterns, a posteriori dietary patterns, overall diet, all-cause mortality, cancer-specific mortality, survival, meta-analysis

## Abstract

The role of overall diet on longevity among cancer survivors (CS) needs further elucidation. We performed a systematic review of the literature and a meta-analysis of related cohort studies published up to October 2022 investigating post-diagnosis a priori (diet quality indices) and a posteriori (data-driven) dietary patterns (DPs) in relation to all-cause and cancer-specific mortality. Pooled hazard ratios (HRs) and 95% confidence intervals (CIs) were estimated using random-effects meta-analyses comparing highest versus lowest categories of adherence to DPs. We assessed heterogeneity and risk of bias in the selected studies. A total of 19 cohort studies with 38,846 adult CS, some assessing various DPs, were included in the meta-analyses. Higher adherence to a priori DPs was associated with lower all-cause mortality by 22% (HR = 0.78, 95% CI: 0.73–0.83, I^2^ = 22.6%) among all CS, by 22% (HR = 0.78, 95% CI: 0.73–0.84, I^2^ = 0%) among breast CS and by 27% (HR = 0.73, 95% CI: 0.62–0.86, I^2^ = 41.4%) among colorectal CS. Higher adherence to a “prudent/healthy” DP was associated with lower all-cause mortality (HR = 0.79, 95% CI: 0.64–0.97 I^2^ = 49.3%), whereas higher adherence to a “western/unhealthy” DP was associated with increased all-cause mortality (HR = 1.48, 95% CI: 1.26–1.74, I^2^ = 0%) among all CS. Results for cancer-specific mortality were less clear. In conclusion, higher adherence to a “healthy” DP, either a priori or a posteriori, was inversely associated with all-cause mortality among CS. A “healthy” overall diet after cancer diagnosis could protect and promote longevity and well-being.

## 1. Introduction

Cancer survivors form a fast-growing segment of the population worldwide. In 2018, 43.8 million people were diagnosed with cancer within the previous five years [1]. Although improvement in cancer survival, observed during the past decades for many cancer sites, is considered a great achievement, cancer survivors have important concerns and face several challenges, such as the late and long-term effects of cancer and its treatment on their survival and quality of life [2,3].

Lifestyle habits and modifications related to a healthy diet and regular physical activity after cancer diagnosis are potentially important behaviors through which cancer survivors could protect and promote their well-being and longevity [4,5].

Several studies among cancer survivors have highlighted that their diet is often characterized by poor dietary habits, unfavorable consumption of specific food groups or nutrients, such as low intake of whole grains and healthy fatty acids, unwanted weight gain and overuse of dietary supplements [6,7,8,9,10]. Furthermore, cancer survivors have consistently expressed their need for additional nutrition guidance and focused dietary advice [11]. Due to a lack of sufficient evidence and shortage of studies conducted among cancer survivors worldwide, currently, dietary recommendations for cancer survivors are the same as those addressed to apparently healthy adults for the primary prevention of cancer, according to the World Cancer Research Fund/American Institute for Cancer Research (WCRF/AICR) Third Expert Report on Diet, Nutrition, Physical Activity, and Cancer [12,13]. More specifically, cancer survivors are encouraged, unless otherwise advised, to eat more whole grains, vegetables, fruits and legumes, to limit consumption of red meat, “fast foods” and processed foods high in fat, starches or sugars and to avoid processed meat, alcohol and sugary drinks. The guidelines stress that although adherence to each of the individual recommendations is endorsed, there is potentially more benefit, if these are treated as an integrated pattern, which, combined with regular physical activity and avoidance of obesity, will have a major impact on cancer risk. 

In that respect, studying the role of dietary patterns, defined as the combinations, quantities and frequencies foods and beverages are habitually consumed, in relation to cancer survivors’ health, is especially appealing since these patterns capture the influence of overall diet on health and well-being, while they also offer an intuitive understanding and an easier interpretation of research findings [14,15,16]. Moreover, the study of dietary patterns, in addition to the study of individual foods and nutrients, is a key element in the process of reviewing the scientific evidence and formulating dietary guidelines for the general population, providing easy to translate, real-life dietary recommendations [17,18]. Two main research approaches have been used for the study of dietary patterns and their association with disease risk [15,16]. The hypothesis-oriented approach uses a priori-defined indices, also called dietary quality indices, to express adherence to a distinct existing dietary pattern, such as the Mediterranean diet, or the level of compliance to formal dietary guidelines, such as guidelines issued by the WCRF/American Institute for Cancer Research (AICR). The data-driven, exploratory approach uses a posteriori-defined dietary patterns, which derive empirically through the application of mathematical/statistical methods, such as principal component and factor analysis, to identify the major dietary pattern/s of a particular study population. Both approaches allow ranking of study participants and quantifying adherence to the specific patterns.

Several systematic reviews, with [19,20,21,22,23,24] or without meta-analyses [25,26,27,28,29], have been conducted so far among cancer survivors, investigating the association between a priori and/or a posteriori dietary patterns, before or after the diagnosis of cancer [20,22] or in both periods [19,21,23,24], in relation to all-cause mortality, cancer-specific mortality or other health outcomes, as well as quality of life. Some of the systematic reviews have focused on studies with survivors from one specific cancer site, such as breast or colorectal cancer survivors, which apparently are the most in numbers so far [21,25,26,27,28], whereas others have included studies with survivors from all cancer sites [19,20,22,24,29].

The majority of findings point out to an inverse association between closer adherence to high-quality diets, as assessed by various a priori dietary patterns or closer adherence to a “prudent/healthy” dietary pattern, as assessed by a posteriori dietary patterns, with all-cause or cancer-specific mortality among cancer survivors [20,21,22,23,24,25,28,29]. In the context of the latest relevant systematic review, a meta-analysis was also performed comparing high versus low adherence to diet quality indices, as assessed by the Healthy Eating Index (HEI) 2005, the HEI 2015 and the Alternative HEI, which found a 23% reduction in overall mortality among breast cancer survivors [23].

Based on the above, the purpose of this study was to synthesize the latest evidence regarding the association of a priori and a posteriori dietary patterns with robust outcome measures, such as total mortality and cancer-specific mortality, expanding our search chronologically, including more databases, implementing robust risk of bias tools and focusing exclusively on the post-diagnosis period, an extremely important period for cancer survivors. 

## 2. Materials and Methods

### 2.1. Search Strategy

The reporting of this systematic review and meta-analysis followed the updated PRISMA guidelines [30]. We focused on specific cancer sites for which there is prior evidence that a diet-related component may be involved in their etiology [12]. We specifically focused on cancer in the mouth, pharynx, larynx, nasopharynx, oesophagus, lung, stomach, pancreas, liver, gallbladder, colorectum (colon), breast, ovaries, endometrium, cervix, prostate, kidney, bladder and skin. The literature search was performed in three electronic databases, MEDLINE, Scopus and Web of Science from January 2000 up to 9 October 2022 (Appendix A). Reference lists of previous meta-analyses and systematic reviews, as well as the identified articles in the present review, were also hand-searched to retrieve any additional relevant articles.

### 2.2. Study Selection

Studies were eligible to be included if: (i) they had a cohort design, prospective or retrospective, (ii) examined the association of a priori or a posteriori dietary pattern or patterns after cancer diagnosis with at least one of the primary endpoints of interest, all-cause mortality and cancer-specific mortality, (iii) the study population consisted of cancer survivors, defined as women and men aged 18 years and older with a diagnosis of a primary cancer (from the time of diagnosis through the remainder of their lives), (iv) the minimum sample size was 100 participants, (v) the length of follow-up was at least six months, and (vi) provided a measure of association, such as Hazard Ratio (HR), and the corresponding 95% confidence intervals (CIs), or sufficient information for their calculation, for the comparison between the highest versus the lowest category of adherence to one a priori or a posteriori dietary pattern. Abstracts and full-texts were independently screened by two authors (ME-S, IB) and disagreement was resolved by consensus with the authors (VB, ES). Following the literature search, studies were screened and the non-relevant ones were excluded: studies with a population not consisting of cancer survivors, cross-sectional studies, case-control studies, studies using as endpoints cancer incidence or quality of life, investigating physical activity and survival, studies that did not use a priori or a posteriori dietary patterns but intake of individual foods, food groups or macro- and micronutrients, studies with changes in dietary intake or low energy reporting, studies investigating the glycemic load index or indices based on biomarkers of diet or inflammation. The search excluded editorials, letters to the editor, comments, conference abstracts, systematic reviews and meta-analyses, and it was limited to English articles.

### 2.3. Data Extraction

All data were extracted in a standard pre-determined format including information on first author, publication year, study location, cancer site, cancer stage (where available), study population, age and sex distribution, follow-up duration, outcome assessed (all-cause mortality and cancer-specific mortality), types of dietary patterns and dietary assessment method used, increments or categories used for the analysis of dietary patterns (i.e., values from quartiles/quintiles used to define the highest category and the lowest category taken as reference), adjustment covariates, and the reported measures of associations (i.e., HR with associated 95% CIs).

We extracted the effect estimates comparing the highest vs. the lowest categories of adherence to a priori dietary patterns assessed in each study (e.g., quartiles, tertiles, good vs. poor adherence, adherers vs. non-adherers, high vs. low score) with the aim to assess the association of adherence to a higher quality diet in relation to all-cause and cancer-specific mortality [20,21,22,31]. In the case which higher adherence to a DP indicated a diet of lower quality, as in the case of the empirical dietary inflammatory pattern (EDIP), the inverse HR comparing those least adhering with those most adhering to that pattern was calculated in order to assess adherence to a higher quality diet. For the a posteriori dietary patterns, based on their characterization by the authors in the primary studies, we created two major groups, the “prudent/healthy” and the “western/unhealthy” group, again comparing the highest vs. the lowest categories of adherence to each pattern. Reported measures of association such as HR, or equivalent estimates, adjusted for the largest number of confounders, were selected and extracted from each study. 

### 2.4. Risk of Bias Assessment of Included Studies

The methodological rigor of the included studies was critically appraised using the ROBINS-I (Risk of Bias In Non-randomized Studies of Interventions) tool, which is proposed for non-randomized studies of interventions/exposures [32]. The ROBINS-I tool comprises seven domains of bias: confounding, selection of participants into the study, classification of exposures, deviations from intended exposures during follow-up, missing data, outcome measurement, and selection of reported result. Each domain is characterized as low, moderate, serious and critical risk of bias, as well as no information (NI), while the overall risk across domains is low only if all domains are characterized as low, moderate if at least one is moderate and high risk of bias if at least one is high. The assessment was performed by two researchers independently, resolving any potential discrepancies by consensus.

### 2.5. Statistical Analysis

The pooled estimate regarding the association of the highest vs. the lowest categories of adherence to post-diagnosis a priori and a posteriori dietary patterns, grouped by cancer site and overall, with each of the outcomes of interest, i.e., all-cause and cancer-specific mortality, was estimated by random effects meta-analysis models to take into account the between-study heterogeneity. The between-studies variance was estimated using the approach by Der Simonian and Laird [33]. Heterogeneity was assessed by the I^2^ statistic, with values >0.50% considered as substantial heterogeneity, and graphically by Galbraith plots [34,35]. Publication bias was assessed by funnel plots. Egger’s test was used to investigate the asymmetry in the case of more than 10 studies. We further applied subgroup analysis among studies by their assessment of the overall risk of bias. A cumulative meta-analysis by year of publication was also performed for all-cause mortality and cancer-specific mortality. A sensitivity analysis with influence plots investigating the impact of a priori dietary patterns on overall mortality and cancer-specific mortality by omitting one study at a time and assessing its effect on the overall estimate was also applied. All analyses were performed using the STATA software and the statistical package metan (version 13.1; StataCorp, College Station, TX, USA) [36,37,38,39].

## 3. Results

### 3.1. Characteristics of Included Studies and Risk of Bias

In this meta-analysis, we included only prospective or retrospective cohort studies and we focused on the dietary patterns that cancer survivors followed after the diagnosis of cancer. Of the 18,964 studies, 41 studies were assessed for eligibility. From those, 22 studies were excluded because of the following reasons: one had a case-control design, one study did not include cancer survivors, five were referring to pre-diagnosis dietary patterns, 10 studies assessed food groups, one study referred to dietary patterns incorporating physical activity, one study included macronutrients, one study included glycemic index, one study included dietary inflammatory index, one study referred to pre-treatment DP (Figure 1). The list of the excluded studies can be found in the Appendix A.

Overall, 19 cohort studies, some of them assessing more than one DP, involving 38,846 cancer survivors, men and women, aged ≥ 18 years old, from four specific cancer sites were included in the meta-analysis. More specifically, 11 studies were conducted among breast cancer survivors, four studies among colorectal cancer survivors, two studies among prostate cancer survivors and two studies among ovarian cancer survivors. The studies on breast cancer survivors included 27,161 women with an average follow-up ranging from 4 to 12 years [40,41,42,43,44,45,46,47,48,49,50]; the studies on colorectal cancer survivors included 4935 men and women with an average follow-up ranging from 5 to 11 years [51,52,53,54]; the studies on prostate cancer survivors included 5464 men with follow-up from 8.7 to 24 years [55,56]; and studies on ovarian cancer survivors included 1286 women with follow-up from 4.4 to 20 years [57,58]. Out of the 19 studies, 16 were conducted in North America, one in Europe, one in Asia and one in Australia. All studies used a validated food frequency questionnaire for the assessment of diet, except one that used a 24-hour recall. The main characteristics of the included studies by cancer site are presented in Table 1.

The dietary patterns reported included 17 a priori distinct dietary patterns and 2 categories of a posteriori dietary patterns characterized by the authors in the published papers as: (a) “prudent/healthy” and (b) “western/unhealthy” dietary patterns. 

With respect to the a priori dietary patterns, these were assessed by the following indices or scores: (1) the Dietary Approaches to Stop Hypertension (DASH) diet, (2) the Healthy Eating Index-2005 (HEI-2005), (3) the Healthy Eating Index-2010 (HEI-2010), (4) the Healthy Eating Index 2015 (HEI-2015), (5) the Alternate Healthy Eating Index (AHEI), (6) the Alternate Healthy Eating Index-2010 (AHEI-2010), (7) the Mediterranean Diet Score (MDS), (8) the alternate Mediterranean Diet Score (aMED), (9) the modified Mediterranean Diet Score (MMDS), (10) the Chinese Food Pagoda-2007 (CHFP-2007), (11) the Chinese Food Pagoda-2016 (CHFP-2016), (12) the Recommended Food Score (RFS), (13) the Diet Quality Index-Revised (DQIR), (14) the American Cancer Society nutrition score (ACS), (15) the Australian Dietary Guidelines Index (DGI), (16) the Empirical dietary inflammatory pattern (EDIP) and (17) the Healthy Nordic Food Index (HNFI). The main characteristics of the indices/scores used to measure adherence to the above mentioned a priori dietary patterns, and more specifically, their characteristic components and short description, are presented in Table 2. 

With respect to a posteriori dietary patterns, the category of “prudent/healthy” dietary pattern was generally characterized by high intakes of fruits and vegetables, whole grains, legumes and fish, while the category of “western/unhealthy” pattern was described in general as a pattern with high intakes of refined grains, red and processed meats, eggs, solid fats, salty snacks and sweets.

The overall risk of bias using the ROBINS-I tool was considered low for 10 of the studies and moderate for nine of the studies (Table 3).

### 3.2. All-Cause Mortality

#### 3.2.1. A Priori DPs and All-Cause Mortality

Τhe results of the random-effects meta-analysis assessing diet quality through a priori dietary patterns with all-cause mortality by cancer site (breast, colorectal, ovarian and prostate cancer survivors) and overall combined are presented in Figure 2.

Overall, a protective association was found between higher adherence to diet quality indices (highest quintile/quartile) versus lower adherence (lowest quintile/quartile) and all-cause mortality, reaching 22% lower risk (HR = 0.78, 95% CI: 0.73–0.83, I^2^ = 23%, *p*-value _for heterogeneity_ = 0.135) among all cancer survivors in 14 unique studies (10 of them assessing more than one a priori pattern). No statistical heterogeneity was observed at the overall effect size or indicated by the Galbraith plot (Appendix A). There was no evidence of asymmetry in the funnel plot, as also indicated by Egger’s test (Appendix A).

Among breast cancer survivors, those who reported higher adherence to these dietary patterns compared to those who reported lower adherence had 22% lower risk of all-cause mortality (HR = 0.78, 95% CI: 0.73–0.84, I^2^ = 0%) in nine unique studies (five of them assessing more than one a priori pattern). Among colorectal cancer survivors, those with higher adherence had 27% lower all-cause mortality (HR = 0.73, 95% CI: 0.62–0.86, I^2^ = 41.4%, *p*-value _for heterogeneity_ = 0.115) compared to those with lower adherence in three unique studies (all assessing more than one a priori pattern). No association was found among ovarian cancer survivors comparing high adherence to a priori DPs vs. low adherence and all-cause mortality (HR = 1.01, 95% CI: 0.67–1.53, I^2^ = 73%, *p*-value _for heterogeneity_ = 0.054) in one study assessing two a priori DPs. With respect to the one study among prostate cancer survivors, the association was inverse, similar to breast and colorectal cancer survivors (HR = 0.78, 95% CI: 0.67–0.90, I^2^ = 0%).

#### 3.2.2. A Posteriori DPs and All-Cause Mortality

The results of the random-effects meta-analyses concerning a posteriori-derived “prudent/healthy” and “western/unhealthy” DPs in relation to all-cause mortality by cancer site (breast, colorectal and prostate cancer survivors) and overall combined are presented in Figure 3 (for the “prudent/healthy” DP) and Figure 4 (for the “western/unhealthy” DP).

A protective association was found between higher adherence to the “prudent/healthy” dietary patterns versus lower adherence and all-cause mortality, reaching 21% lower risk (HR = 0.79, 95% CI: 0.64–0.97, I^2^ = 49.3%, *p*-value _for heterogeneity_ = 0.079; Figure 3) among all cancer survivors. Evidence of heterogeneity was substantial, but the number of studies was small (n = 6). Breast cancer survivors who reported higher adherence in comparison to those who reported lower adherence to a “prudent/healthy” dietary pattern had a 31% lower risk of all-cause mortality (HR = 0.69, 95% CI: 0.51–0.93, I^2^ = 9.3%, *p*-value _for heterogeneity_ = 0.294). No heterogeneity was observed regarding the two studies included in the meta-analysis among breast cancer survivors. Similarly, adherence to a “prudent/healthy” dietary pattern was negatively associated with all-cause mortality among colorectal cancer survivors, but the association was not statistically significant (HR = 0.93, 95% CI: 0.66–1.30, I^2^ = 65.8%, *p*-value _for heterogeneity_ = 0.054), and although substantial heterogeneity was observed, this was based on few studies (n = 3). The one study on prostate cancer survivors reported 36% lower risk (HR = 0.64, 95% CI: 0.44–0.93) comparing highest with lowest adherence to a “prudent/healthy” dietary pattern.

Regarding the “western/unhealthy” dietary pattern, among cancer survivors from all cancer sites in six studies, higher adherence to this pattern had 48% higher risk of all-cause mortality compared to lower adherence (HR = 1.48, 95% CI: 1.26–1.74, I^2^ = 0%; Figure 4) with no evidence of heterogeneity. In the two studies among breast cancer survivors who reported highest adherence in comparison to lowest adherence, a 53% higher risk of all-cause mortality was observed (HR = 1.53, 95% CI: 1.12–2.09, I^2^ = 0%). In three studies with colorectal cancer survivors, the highest adherence to a “western/unhealthy” dietary pattern compared to lowest adherence had a 47% higher risk of all-cause mortality (HR = 1.47, 95% CI: 1.05–2.05, I^2^ = 52.9%, *p*-value _for heterogeneity_ = 0.120). Only one study reported results among prostate cancer survivors where the highest level of adherence to a “western” dietary pattern had a 67% higher risk of all-cause mortality compared to the lowest (HR = 1.67, 95% CI: 1.16–2.41).

### 3.3. Cancer-Specific Mortality

#### 3.3.1. A Priori DPs and Cancer-Specific Mortality

Although an inverse association was noted between higher adherence to a priori dietary patterns and cancer-specific mortality among cancer survivors from all sites, this did not reach statistical significance (HR = 0.91, 95% CI: 0.82–1.01, I^2^ = 47.2%, *p*-value _for heterogeneity_ = 0.003; 13 studies; Appendix A). There was no evidence of asymmetry in the funnel plot, as also indicated by Egger’s test (Appendix A).

Among breast cancer survivors, higher adherence to a diet quality index was associated with lower breast cancer mortality, but again the association was not statistically significant (HR = 0.90, 95% CI: 0.78–1.03, I^2^ = 48.9%, *p*-value _for heterogeneity_ = 0.010; seven studies; Appendix A). We observed moderate heterogeneity and no evidence of asymmetry in the funnel plot. Results from Egger’s test indicated that there was no evidence of the small study effect in breast cancer studies (Appendix A).

On the other hand, among colorectal cancer survivors, colorectal cancer mortality was statistically significantly lower, by 33%, among those with high adherence to a priori dietary patterns compared to lower (HR = 0.67, 95% CI: 0.50–0.88, I^2^ = 26.1%, *p*-value _for heterogeneity_ = 0.248; Appendix A). There was no evidence of asymmetry or heterogeneity, but the number of studies was small (two unique studies assessing more than one pattern; Appendix A). Among ovarian cancer survivors in two unique studies assessing more than one pattern, high adherence to a priori DPs was not statistically associated with cancer mortality (HR = 1.09, 95% CI: 0.92–1.28, I^2^ = 0%, *p*-value = 0.434; Appendix A). The one study among prostate cancer survivors reported no association (HR = 0.98; 95% CI: 0.75–1.29).

#### 3.3.2. A Posteriori DPs and Cancer-Specific Mortality

With respect to adherence to a “prudent/healthy” dietary pattern, an inverse association was found among cancer survivors from all sites (HR = 0.76, 95% CI: 0.58–0.99, I^2^ = 0%, *p*-value _for heterogeneity_ = 0.451; Appendix A) among five studies. Among colorectal cancer survivors, in two studies, higher adherence to a “prudent/healthy” dietary pattern was associated with 36% lower colorectal cancer mortality (HR = 0.64, 95% CI: 0.43–0.95, I^2^ = 0%, *p*-value _for heterogeneity_ = 0.848). No association was evident among breast cancer survivors and prostate cancer survivors (Appendix A).

Regarding the association between the “western/unhealthy” dietary pattern and cancer-specific mortality, higher adherence was associated with 41% higher cancer mortality among cancer survivors from all sites (HR = 1.41, 95% CI: 1.06–1.88, I^2^ = 0%, *p*-value _for heterogeneity_ = 0.408; five studies; Appendix A), with 69% higher colorectal cancer mortality (HR = 1.69, 95% CI: 1.09–2.64, I^2^ = 0%, *p*-value _for heterogeneity_ = 0.938; two studies) among colorectal cancer survivors, whereas results were not statistically significant for breast cancer survivors (HR = 1.08, 95% CI: 0.72–1.62, I^2^ = 0%, *p*-value _for heterogeneity_ = 0.688; two studies).

### 3.4. Sensitivity Analysis

In sensitivity analysis, the pooled estimate for all-cause mortality in relation to a priori DPs decreased by 1.3%, when the studies by Kim et al., 2011 [42], by McCullough et al., 2016 [46] and by Sasamoto et al. 2022 [58] were omitted, and increased by a range between 1.3% to 2.6% when the studies by Izano et al., 2013 [44], Ratjen et al., 2017 [53] and Guinter et al., 2018 [54] were omitted (Appendix A). The pooled estimate for breast cancer mortality regarding a priori DPs, decreased by 3% when the study by Kim et al., 2011 [42] and McCullough et al., 2016 [46] were omitted and by 2% when the study by Izano et al., 2013 [44] was omitted, and increased by 2% when the study by Wang et al., 2020 [49] was omitted (Appendix A).

Cumulative meta-analysis for a priori dietary patterns and all-cause mortality by year of publication indicated a change in the estimates of HR, with the HR increasing and moving away from the null (HR_2011_ = 0.85, 95% CI: 0.62–1.16; HR_2022_ = 0.78, 95% CI: 0.73–0.83) (Appendix A). Regarding breast cancer mortality, cumulative meta-analysis by year of publication indicated the opposite change, since the overall estimate decreased in magnitude among studies published between 2011 and 2021 (HR_2011_ = 0.12, 95% CI: 0.02–0.84; HR_2021_ = 0.89, 95% CI: 0.77–1.03) (Appendix A).

### 3.5. Subgroup Analysis

A subgroup analysis for the association between a priori DPs and overall mortality by risk of bias showed a similar association both among the nine studies with moderate risk (HR = 0.72, 95% CI: 0.63–0.82, I^2^ = 46%, *p*-value _for heterogeneity_ = 0.035) and among the 10 studies with low risk of bias (HR = 0.81, 95% CI: 0.76-0.86, I^2^ = 0%) (Appendix A). Regarding the association of a priori DPs and breast cancer mortality, the subgroup analysis revealed an inverse association only among studies with moderate risk of bias (HR = 0.67, 95% CI: 0.53–0.83, I^2^ = 24.3%, *p*-value _for heterogeneity_ = 0.259) (Appendix A).

## 4. Discussion

In this systematic review and meta-analysis of cohort studies focusing on the role of post-diagnosis dietary patterns in survival among cancer survivors, greater adherence to diets of higher quality, as assessed by a priori dietary patterns, or greater adherence to a “prudent/healthy” a posteriori dietary pattern, were associated with a significant reduction in all-cause mortality. On the other hand, greater adherence to a “western/unhealthy” a posteriori dietary pattern was associated with a significant increase in all-cause mortality. Among cancer survivors of different sites, the inverse association observed between adherence to a priori dietary patterns and all-cause mortality was more pronounced for colorectal cancer survivors compared to breast cancer survivors or survivors from other sites. Adherence to a “western/unhealthy” dietary pattern was associated with higher all-cause mortality in cancer survivors from all cancer sites, whereas adherence to a “prudent/healthy” dietary pattern was associated with lower all-cause mortality mainly among breast cancer survivors.

With respect to cancer-specific mortality, findings were less clear and were based on few studies; thus, they should be interpreted with caution. Nevertheless, higher adherence to a “western/unhealthy” dietary pattern was associated with increased cancer mortality and higher adherence to a “prudent/healthy” dietary pattern with decreased cancer mortality among all cancer survivors and among colorectal cancer survivors. Cumulative meta-analysis by year of publication revealed stronger inverse associations between a priori dietary patterns and all-cause mortality among cancer survivors of all sites during the more recent years. This could be partly attributed to updates and time-dependent improvements in the characteristics of a priori dietary patterns used in these studies, such as the Healthy Eating Index [22].

Our findings are, in general, in agreement with previous systematic reviews and meta-analyses in which adherence to various a priori dietary patterns was associated with longer survival among cancer survivors [19,20,21,22,23,24]. The a priori dietary patterns investigated in this meta-analysis have differences in terms of their characteristic components, construction of scores and final scoring, as shown in Table 2. Yet, their association with all-cause mortality was evident when they were meta-analyzed together. A careful inspection of their individual components can lead to the observation that all of them reflect core constituents of a “healthy” diet [15]. This “healthy” diet is mostly plant-based, characterized by a high consumption of vegetables, fruits, whole grains, nuts, legumes and preference of plant oils, and less animal-based, characterized by low consumption of red and/or processed meat and moderate dairy consumption. Consumption of foods with low salt content and limited added sugars are also common characteristics of these dietary patterns. The same applies for the “prudent” and “western” a posteriori-derived dietary patterns, which all consist of food group combinations that have been associated with better health for the former, and with worse health for the latter. Thus, the “prudent/healthy” dietary patterns are usually characterized by high intake of fruits, vegetables, whole grains, legumes, fish and low intake of red and processed meat, whereas the ‘‘unhealthy/western’’ dietary patterns are characterized by high intakes of animal-based products, processed meats, refined grains, sweets and desserts, sweetened beverages and salty snacks [52,54,56]. The plausible biological pathways through which the above mentioned a priori and a posteriori “prudent” dietary patterns, may exert their beneficial effects on health, overall and cancer-specific survival, could be attributed to the anti-inflammatory, antithrombotic, antioxidative and antioncogenic properties of vitamins, antioxidants, phytochemicals, potassium, folate, minerals, fiber and healthy fatty acids, constituents abundant in their characteristic food groups [12,76]. On the contrary, the a posteriori “western/unhealthy” dietary patterns may exert their detrimental effects through their high content in saturated and trans fatty acids, added sugars, salt and refined grains and low content in fiber, vitamins, minerals and antioxidants [76].

Compared to previous meta-analyses, our meta-analysis includes solely observational studies, cohort studies in particular, and focuses exclusively on the post-diagnosis period. Although intervention studies can provide the strongest and most direct epidemiologic evidence for the existence of a cause-effect association, they have unique challenges in terms of feasibility, compliance and cost in the context of nutritional epidemiology and especially in the context of investigating “hard” primary outcomes such as mortality [77]. Most randomized controlled trials (RCTs) conducted among cancer survivors investigating the role of diet in survivors’ prognosis so far have used indices of quality of life as primary outcomes and not survival [23]. Carefully conducted observational studies, with low risk of bias, like most of the cohorts included in this meta-analysis, can provide reliable and reproducible evidence on diet and health relationships, whereas well-designed RCTs, of course, can contribute substantially [78]. The focus on the post-diagnosis period in this meta-analysis was deemed necessary to highlight the relative importance of adhering to specific dietary patterns after cancer diagnosis and be able to proceed to specific dietary recommendations for this period. Also, we decided to meta-analyze different a priori indices in relation to all-cause and cancer-specific mortality, as was done in a previous meta-analysis [20], with the rationale that all were constructed based on adherence to a dietary pattern characterized by healthy food groups and/or dietary recommendations to achieve better health.

Limitations of our analysis include the relatively small number of cohort studies conducted during the post-diagnosis period among cancer survivors. Also, the available studies to conduct meta-analysis were limited to breast and colorectal cancer survivors, whereas those on prostate and ovarian cancer survivors were too few to make reliable conclusions. Studies among cancer survivors from other sites with some evidence of diet involvement in their etiology during the post-diagnosis period were not found. The meta-analysis of different a priori dietary patterns can be considered a limitation due to their differences in construction and heterogeneous nature and the inability to attribute and translate the findings to a single dietary pattern. On the other hand, an inherent strength of our study is the investigation of dietary patterns instead of individual food groups in relation to health and disease, which has several advantages, such as better capture of the nutrient-nutrient interactions and food synergies between the individual components of the patterns consumed. Another strength is the search in three databases and the extent of the search until October 2022 with the inclusion of seven additional cohort studies in relation to previous meta-analyses with respect to post-diagnosis a priori or a posteriori dietary patterns among cancer survivors diagnosed with cancers with a probable nutrition-related aspect in their etiology [19,20,21,22,23,24]. Although inherent methodological limitations of observational studies should be considered when interpreting this data, such as residual confounding (e.g by cancer treatment or pre-diagnosis exposures), selection bias or measurement error (e.g self-reported assessment of diet only at one point in time), cohort studies included in this meta-analysis had low to moderate risk of bias assessed by the ROBINS-I tool and results were similar both among low and moderate risk of bias studies. The existence of publication bias and heterogeneity were unlikely based on the results from specific tests and plots.

In conclusion, this systematic review and meta-analysis supports the beneficial role of “healthy” dietary patterns, either a priori or a posteriori, during the post-diagnosis period, in relation to all-cause mortality among cancer survivors. Continuous research on the role of dietary patterns after cancer diagnosis is needed in order to confirm the important role of overall diet and issue evidence-based dietary recommendations that will preserve and promote the health and well-being of cancer survivors.

## Figures and Tables

**Figure 1 nutrients-15-03860-f001:**
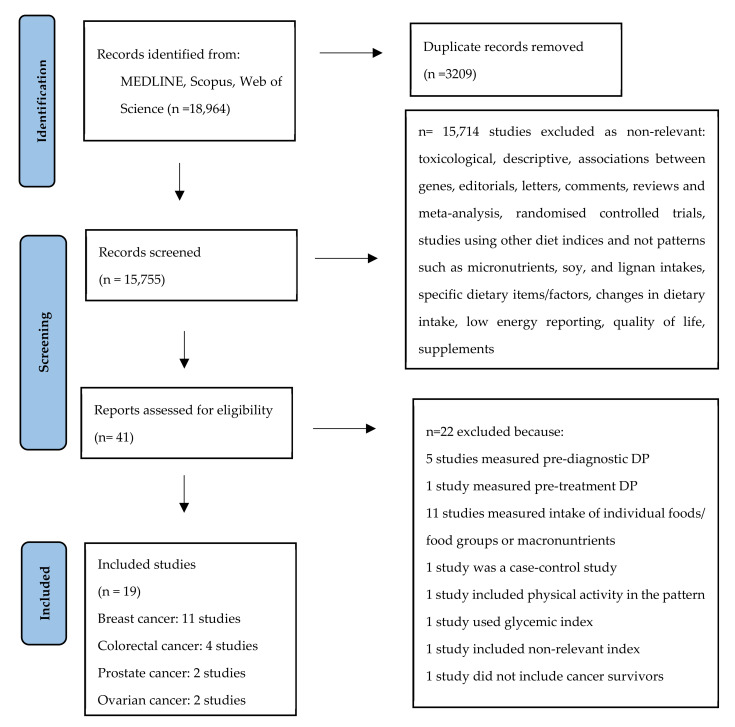
Flow chart of the literature search process.

**Figure 2 nutrients-15-03860-f002:**
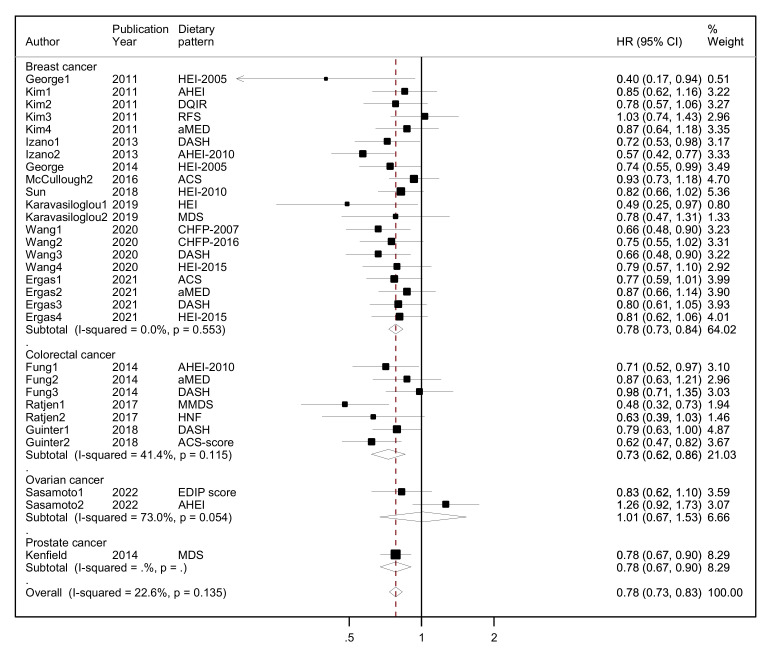
Forest plot showing the association between highest versus lowest adherence to a priori dietary patterns with all-cause mortality by cancer site and overall, among cancer survivors. Abbreviations: CHFP-2007: Chinese Food Pagoda-2007, CHFP-2016: Chinese Food Pagoda-2016, MDS: Mediterranean Diet Score, HEI-2015: Healthy Eating Index-2015, HEI-2005: Healthy Eating Index-2005, HEI-2010: Healthy Eating Index-2010, aMED, altMed: alternate Mediterranean Diet Score, MMDS: Modified Mediterranean Diet Score, HNFI: Healthy Nordic Food Index, RFS: Recommended Food Score, DASH: Dietary Approaches to Stop Hypertension, EDIP: Empirical dietary inflammatory pattern, ACS: American Cancer Society, DQIR: Diet Quality Index-Revised, AHEI: Alternate Healthy Eating Index, AHEI-2010: Alternate Healthy Eating Index-2010, DGI: Australian Dietary Guideline Index.

**Figure 3 nutrients-15-03860-f003:**
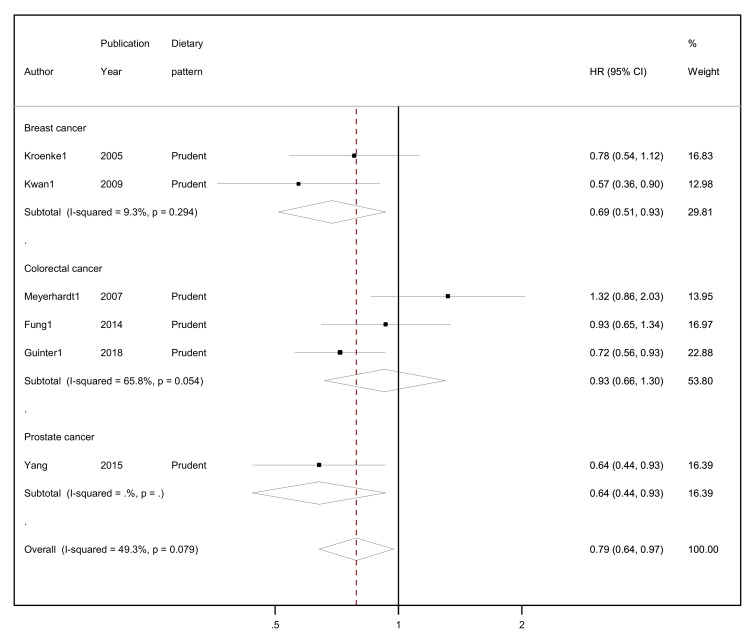
Forest plot showing the association between highest versus lowest adherence to “prudent/healthy” dietary patterns with all-cause mortality by cancer site and overall, among cancer survivors.

**Figure 4 nutrients-15-03860-f004:**
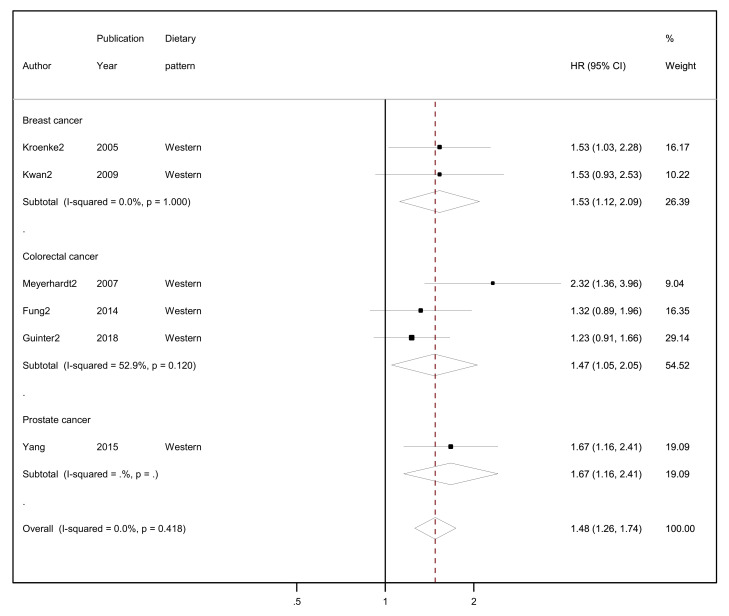
Forest plot showing the association between highest versus lowest adherence to “western/unhealthy” dietary patterns with all-cause mortality by cancer site and overall, among cancer survivors.

**Table 1 nutrients-15-03860-t001:** Main characteristics of the cohort studies included in this meta-analysis by cancer survivor site.

Author, Year, Location	Cancer Site	Outcome of Interest	N, Age, Gender, Follow-Up Duration	Dietary Patterns (DPs) Used	Dietary Assessment Tools	Adjustment Factors
**Breast cancer (BC)**						
Kroenke et al., 2005, USA [40]	Breast cancer	all-cause mortality	2619 women, age at diagnosis by DP: Prudent = 58 years, Western = 58 years, median follow-up = 9 years	Prudent, Western	validated FFQ	Age, BMI, energy intake, smoking, PA, diet missing in 1986, 1990, 1994, age at menarche, OCU, MS, HRT, age at menopause, stage, tamoxifen, chemotherapy
Kwan et al., 2009, USA [41]	Breast cancer	all-cause mortality, BC mortality, mortality from causes other than BC, recurrence	1901 women, age at diagnosis by DP: Prudent = 58.4 years, Western = 58.9 years, mean follow-up = 4.2 years	Prudent, Western	validated FFQ	Age, energy intake, race, BMI, PA, smoking, MS, weight change, stage, HRS, treatment
Kim et al., 2011, USA [42]	Breast cancer	all-cause mortality, BC mortality, mortality from causes other than BC	2729 women, Age at baseline = 30–55 years, follow-up until 30 June 2004	AHEI, DQIR, RFS, aMED	validated FFQ	Time since diagnosis, age, alcohol intake (for RFS only), energy, multivitamin use, BMI, weight change, OCU, smoking, PA, stage, treatment, age at first birth and parity, MS, HRT
George et al., 2011, USA [43]	Breast cancer	all-cause mortality, BC mortality	670 women, age at diagnosis = 18–64 years, mean follow-up = 6 years	HEI-2005	validated FFQ	Energy intake, PA, race, stage, tamoxifen use, BMI
Izano et al., 2013, USA [44]	Breast cancer	BC mortality, death from causes other than BC	4103 women, age at baseline = 30–55 years, median follow-up = 9.3 years	DASH, AHEI-2010	validated FFQ	Age, energy intake, BMI, BMI change, age at first birth and parity, OCU, MS, HRT, smoking, stage, treatment, PA
George et al., 2014, USA [45]	Breast cancer	all-cause mortality, BC mortality, death from causes other than BC	2317 women, age at screening for WHI: HEI-2005 Q1:63.6 years, Q2: 63.6 years, Q3: 63.4 years, Q4:63.9 years, median follow-up = 9.6 years	HEI-2005	validated FFQ	Age, WHI component, ethnicity, income, education, stage, ER, PR, time since diagnosis, energy intake, PA, alcohol consumption, HRT
McCullough et al., 2016, USA [46]	Breast cancer	all-cause mortality, BC mortality, death from causes other than BC	2152 women, age at diagnosis = 70.7 years, mean follow-up = 9.9 years	ACS score	validated FFQ	Age, year of diagnosis, stage, grade, ES, PR, initial treatment and the following assessed at the time of FFQ completion: BMI, smoking, PA and energy intake (Q)
Sun et al., 2018, USA [47]	Breast cancer	all-cause mortality, BC mortality	2295 women, age at diagnosis = 66 years, mean follow-up = 12 years	HEI-2010	validated FFQ	Age, energy intake, alcohol, smoking, PA, race, ethnicity, education, SES, stage, ER, PR, time from diagnosis to dietary intake assessment, postmenopausal hormone therapy use, BMI
Karavasiloglou et al., 2019, USA [48]	Breast cancer	all-cause mortality	230 (110 breast cancer survivors and 120 gynecological cancer), age at diagnosis = 53.7 years, median follow-up = 8.6 years	HEI-2005, MDS	24-h recall	Time between cancer diagnosis and completion of the NHANES III questionnaire, SES, marital status, BMI, PA, smoking, self-reported prevalent chronic diseases, daily energy intake, history of menopausal HRT, alcohol intake in the analyses for the HEI but not for the MDS
Wang et al., 2020, China [49]	Breast cancer	all-cause mortality, BC mortality, breast cancer-specific events	3450 women, age at diagnosis = 59 years, follow-up duration = 10 years	CHFP-2007, CHFP-2016, DASH, HEI-2015	validated FFQ	Age, intervals between diagnosis and 60-month survey, energy intake, income, education, marital status, MS, BMI, PA, ER, PR, HER2, TNM stages, comorbidity, chemotherapy, radiotherapy and immunotherapy
Ergas et al., 2021, USA [50]	Breast cancer	all-cause mortality, BC mortality, BC recurrence	3660 women, age at diagnosis = 59.7 years, follow-up duration = 40 years	ACS, aMED, DASH, HEI-2015	validated FFQ	Age, total energy intake, race, ethnicity, education, menopausal status, PA, smoking, stage, ER, PR, HER2, BMI, surgery type, chemotherapy, radiation and hormonal therapies
**Colorectal cancer (CRC)**						
Meyerhardt et al., 2007, USA and Canada [51]	Colorectal cancer (stage III)	all-cause mortality, cancer recurrence	1009 men, women, median age at diagnosis by DP: Prudent = 61 years, Western = 62 years median follow-up = 5.3 years	Prudent, Western	validated SFFQ	Age, sex, depth of invasion, positive lymph nodes, clinical perforation at surgery, bowel obstruction at surgery, baseline performance status, treatment, weight change, time-varying BMI, PA, total calories
Fung et al., 2014, USA [52]	Colorectal cancer (stage I-III)	overall survival, CRC specific mortality	1201 women, median age at diagnosis = 66.5 years, median follow-up = 11.2 years	DASH, aMED, AHEI-2010, Prudent, Western	SFFQ	Age, PA, BMI, weight change, grade, chemotherapy, smoking, energy intake, stage, tumor site and date of CRC diagnosis
Ratjen et al., 2017, Germany [53]	Colorectal cancer	all-cause mortality	1404 patients, age at diet assessment = 69 years, median age at diagnosis = 62 years, median follow-up = 7 years	MMDS, HNFI	validated, web-based SFFQ	Age, sex, BMI, PA, survival time from CRC diagnosis until diet assessment, tumor location, occurrence at metastases, occurrence of other tumor, chemotherapy, smoking, energy intake, time x age, time x BMI, and time x metastases
Guinter et al., 2018, USA [54]	Colorectal cancer	all-cause mortality, CRC-specific mortality	Post-diagnosis sample: 1321 men, women, age at baseline = 64.6 years, age at diagnosis = 70.6 years, mean follow-up = 6.4 years	DASH, ACS score, Prudent, Western	validated FFQ	Age, year of diagnosis, sex, stage, total calorie intake, BMI, education, smoking, weight change since 1992, treatment
**Ovarian Cancer (OC)**						
Al Ramadhani et al., 2020, Australia [57]	Ovarian cancer	OC mortality	503 women, ages = 18–79 years, mean follow-up = 4.4 years	HEI-2010, AHEI-2010, ACS score, DGI	validated FFQ	Age, energy intake, smoking at 12 months (never/former/current), and FIGO stage, and stratified by PA at 12 months
Sasamoto et al., 2022, USA [58]	Ovarian cancer	all-cause mortality, OC mortality	783 women, median age at diagnosis = 62 years, follow-up of NHS II = 6 years, follow-up of NHS = 20 years	EDIP, AHEI-2010	validated FFQ	Age, year at diagnosis, histology, stage, smoking, BMI, energy intake, NSAID use
**Prostate Cancer (PC)**						
Kenfield et al., 2014, USA [55]	Prostate cancer	all-cause mortality	4538 men, age in 1990 by diet score 0 to 3:52.6 years, 4 to 5: 54.3 years, 6 to 9: 55.3 years, follow-up until 31 January 2010	MDS	validated SFFQ	Age, time period, time since diagnosis to FFQ, energy intake, BMI, vigorous PA, smoking, stage, Gleason score, treatment, race, height, history of diabetes mellitus, family history of PC, multivitamin use
Yang et al., 2015, USA [56]	Prostate cancer (non-metastatic)	all-cause mortality	926 men, age at diagnosis = 68.6 years, median follow-up = 9.9 years	Prudent Western	validated FFQ	Age, energy intake, BMI, smoking, vigorous PA, Gleason score, stage, prostate-specific antigen level, time interval between diagnosis and FFQ completion, initial treatment after diagnosis, family history of PC

Abbreviations: DP: Dietary patterns, BC: Breast Cancer, CRC: Colorectal Cancer, PC: Prostate cancer, OC: Ovarian cancer, FFQ: Food Frequency Questionnaire, SFFQ: semi quantitative food frequency questionnaires, CHFP-2007: Chinese Food Pagoda-2007, CHFP-2016: Chinese Food Pagoda-2016, MDS: Mediterranean Diet Score, HEI-2015: Healthy Eating Index-2015, HEI-2005: Healthy Eating Index-2005, HEI-2010: Healthy Eating Index-2010, aMED, altMed: alternate Mediterranean Diet Score, MMDS: Modified Mediterranean Diet Score, HNFI: Healthy Nordic Food Index, RFS: Recommended Food Score, DASH: Dietary Approaches to Stop Hypertension, EDIP: Empirical dietary inflammatory pattern, ACS: American Cancer Society, DQIR: Diet Quality Index-Revised, AHEI: Alternate Healthy Eating Index, AHEI-2010: Alternate Healthy Eating Index-2010, DGI: Australian Dietary Guideline Index, ER: Estrogen Receptor, PR: Progesterone Receptor, HER2:Human epidermal growth factor receptor-2, TNM: Tumor Node and Metastasis staging system, OM: overall mortality, cMRM: combined mortality, metastasis or recurrence, BMI: Body Mass Index, PA: Physical Activity, HRT: Hormone replacement therapy, MS: Menopausal status, WHI component: Women’s Health Initiative component, HRS: Hormone receptor status, OCU: oral contraceptive use, NHANES III: National Health and Nutrition Examination Survey, FIGO: Fédération Internationale de Gynécologie et d’Obstétrique (International Federation of Obstetrics and Gynecologic), NSAID: Nonsteroidal anti-inflammatory drug, Q: quartile, SES: socioeconomic status, N: study population.

**Table 2 nutrients-15-03860-t002:** Main characteristics of a priori dietary patterns used in the studies included in this meta-analysis.

A Priori Dietary Pattern	Main Characteristics of the Pattern *(Reference)*	Papers in this Meta-Analysis
Healthy Eating Index (HEI-2005)	-Measures adherence to the 2005 Dietary Guidelines for Americans -Includes 12 components that represent the major food groups found in MyPyramid -Diets that meet the least restrictive of the food-group recommendations (expressed on a per 1000 calorie basis) receive maximum scores for the nine adequacy components of the index: total vegetables (5 points), dark green and orange vegetables and legumes (5 points), total fruit (includes 100% juice) (5 points), whole fruit (5 points), total grains (5 points), whole grains (5 points), milk (10 points), meat and beans (10 points) and oils (10 points). In moderation: saturated fat, sodium, calories from solid fats, alcoholic beverages and added sugars (which serve as a proxy for discretionary calories) *Ref: Guenther PM, Reedy J, Krebs-Smith SM. Development of the Healthy Eating Index-2005. J. Am. Diet. Assoc. 2008;108:1896–901* [59]	George et al., 2011 [43]; George et al., 2014 [45]; Karavasiloglou et al., 2019 [48]
Alternate Healthy Eating Index (AHEI)	-Designed to target food choices associated with reduced risk for chronic diseases -Based on 1992 Food Guide Pyramid and 1995 Dietary Guidelines for Americans -Consists of 9 components: vegetables (no potatoes included), fruits, nuts and soy, fiber cereals, white/red meat ratio, trans fatty acids, polyunsaturated/saturated fatty acids ratio, alcohol -Each component takes 0 to 10 points (zero to max adherence). Multivitamin use duration was scored dichotomously: 7.5 points for ≥5 y regular use, 2.5 points for all others, to avoid overly emphasizing this component. -Score ranges from 2.5 (lower) to 87.5 (highest) *Ref: McCullough ML, Feskanich D, Stampfer MJ,* et al. *Diet quality and major chronic disease risk in men and women: moving toward improved dietary guidance. Am. J. Clin. Nutr. 2002; 76:1261–1271.* [60]	Kim et al., 2011 [42]; Sasamoto et al., 2022 [58]
Healthy Eating Index-2010 (HEI-2010)	-The HEI-2010 retains several features of the HEI-2005: (1) Consists of 12 components, including 9 adequacy components: total fruit, whole fruit, total vegetables, greens and beans, whole grains, dairy, total protein foods, seafood and plant proteins, fatty acids and 3 moderation components: grains, sodium and empty calories; (2) Density approach to setting standards: e.g., per 1000 calories. Least--restrictive standards, i.e., those that are easiest to achieve among recommendations that vary by energy level, sex and/or age. -Changes to the index include: (1) Dark green and orange vegetables and legumes replaced greens and beans; (2) plant proteins and seafood have been added; (3) a ratio of poly- and mono-unsaturated to saturated fatty acids replaced oils and saturated fat; and (4) a moderation component, refined grains, replaced the adequacy component total grains *Ref: Guenther PM, Casavale KO, Reedy J, Kirkpatrick SI, Hiza HA, Kuczynski KJ, Kahle LL, Krebs-Smith SM. Update of the Healthy Eating Index: HEI-2010. J Acad Nutr Diet. 2013 Apr;113(4):569-80. Erratum in: J. Acad. Nutr. Diet. 2016 Jan;116(1):170. PMID: 23415502; PMCID: PMC3810369.* [61]	Sun et al., 2018 [47]; Al Ramadhani et al., 2020 [57]
Alternate Healthy Eating Index-2010 (AHEI-2010)	-Components were chosen based on their association with chronic diseases -Points were awarded for higher consumption of vegetables (excluding potatoes), whole grains, nuts, whole fruit and legumes, long chain omega-3 fatty acids and polyunsaturated fatty acids -Lower consumption for red/processed meat, sugar-sweetened beverages, sodium, trans fatty acids and moderate alcohol intake -Each food group could take a value from 0 to 10 points (maximum overall score: 110 points). *Ref:**Chiuve SE, Fung TT, Rimm EB, Hu FB, McCullough ML, Wang M, Stampfer MJ, Willett WC. Alternative dietary indices both strongly predict risk of chronic disease. J. Nutr. 2012 Jun;142(6):1009-18.* [62]	Izano et al., 2013 [44]; Fung et al., 2014 [52]; Al Ramadhani et al., 2020 [57]
Healthy Eating Index (HEI-2015)	-Consists of 13 components: total vegetables, greens and beans, total fruits, whole fruits, dairy, seafood and plant protein, refined grains, total protein, added sugars, fatty acids, saturated fatty acids and sodium and -Each component scored a maximum of 10 points; for components divided into two, each subcomponent was allocated 5 points. -For the HEI-2015, only the 1200 to 2400 kcal patterns were used (compared with the range of 1000 to 3200 kcal, used for some components in prior versions). *Ref: Kirkpatrick SI, Reedy J, Krebs-Smith SM, Pannucci TE, Subar AF, Wilson MM, Lerman JL, Tooze JA. Applications of the Healthy Eating Index for Surveillance, Epidemiology, and Intervention Research: Considerations and Caveats. J. Acad. Nutr. Diet. 2018 Sep;118(9):1603-1621.* [63]	Wang et al., 2020 [49]; Ergas et al., 2021 [50]
Diet Quality Index-Revised (DQIR)	-Consists of 10 components: vegetables, fruits, grains, total fat, saturated fat, cholesterol, iron, calcium, diet diversity, added fat and sugar moderation -Each component scores from 0 to 10 based on the recommended range of intakes. -The maximum possible score is 100. *Ref: Haines PS, Siega-Riz AM, Popkin BM. The Diet Quality Index revised: a measurement instrument for populations. J. Am. Diet. Assoc. 1999 Jun;99(6):697-704.* [64]	Kim et al., 2011 [42]
Recommended Food Score (RFS)	-Consists of 23 components: apples/pears; oranges/grapefruit; cantaloupes; orange/grapefruit juice; grapefruit; other fruit juices; dried beans; tomatoes; broccoli; spinach; mustard, turnip/collard greens; carrots/mixed vegetables with carrots; green salad; sweet potatoes, yams/other potatoes; baked or stewed chicken or turkey; baked or broiled fish; dark breads; cornbread, tortillas and grits; high-fiber cereals; cooked cereals; 2% milk and beverages with 2% milk; and 1% milk or skimmed milk. -The score is calculated by summing each of these 23 items that was consumed at least once a week, for a maximum score of 23. *Ref: Mai V, Kant AK, Flood A, Lacey JV Jr, Schairer C, Schatzkin A. Diet quality and subsequent cancer incidence and mortality in a prospective cohort of women. Int. J. Epidemiol. 2005 Feb;34(1):54-60.* [65]	Kim et al., 2011 [42]
Mediterranean Diet Score (MDS)	-Consists of 9 components: vegetables, fruit and nuts, legumes, cereals, fish and seafood, meat and meat products, dairy, ratio of monounsaturated to saturated fatty acids and alcohol intake. -Participants with consumption of legumes, vegetables, fruit and nuts, cereals, and fish and seafood above the sex-specific population median received 1 point (per component), whereas consumption of meat and meat products, dairy products and a ratio of monounsaturated to saturated fats lower than the population median received 1 point (per component). Ethanol consumption of 5–25 g/day received 1 point and 0 points otherwise. -The MDS takes values from 0 to 9. *Ref: Trichopoulou A, Costacou T, Bamia C, Trichopoulos D. Adherence to a Mediterranean diet and survival in a Greek population. N. Engl. J. Med. 2003 Jun 26;348(26):2599-608.* [66]	Kenfield et al., 2014 [55]; Karavasiloglou et al., 2019 [48]
Alternate Mediterranean Diet Score (aMED)	-This score adapts the principles of the traditional Mediterranean diet to non-Mediterranean countries -The Mediterranean diet score was changed as follows: potato products were excluded from the vegetable group, fruits and nuts formed 2 separate groups, the grain group included only whole-grain products, the meat group included only red and processed meat. Participants received 1 point when they consumed > than the median intake of vegetables, legumes, fruits, nuts, whole grains, fish and monounsaturated/saturated fat ratio, and otherwise received 0 points for the particular food group. Participants received 1 point for consuming less than the median intake of meat and dairy. For ethanol: 1 point was assigned for intake between 5 and 15 g/d. -The score ranges from 0 to 9. *Ref: Fung TT, McCullough ML, Newby PK, Manson JE, Meigs JB, Rifai N, Willett WC, Hu FB. Diet-quality scores and plasma concentrations of markers of inflammation and endothelial dysfunction. American Journal of Clinical Nutrition. 2005;82:163–173.* [67]	Kim et al., 2011 [42]; Fung et al., 2014 [52]; Ergas et al., 2021 [50]
Modified Mediterranean Diet Score (MMDS)	-This score modified the original MDS by Trichopoulou et al. to be applied to the non-Mediterranean countries -Consists of 9 components: vegetables, legumes, fruit and nuts, cereals, fish and seafood, meat and meat products, dairy products, unsaturated/saturated fatty acids ratio and alcohol. -A value of 1 for an intake at or above the sex-specific median of 5 components (vegetables, fruits and nuts, legumes, cereals and fish) and for an intake below the sex-specific median of 2 components (meat products and dairy products); otherwise, a value of 0 was assigned. For ethanol, a value of 1 was assigned to men who consumed between 10 and 50 g/d and to women who consumed between 5 and 25 g/d; otherwise, a value of 0 was assigned. -For fat intake, the ratio of unsaturated lipids (monounsaturated and polyunsaturated lipids) to saturated lipids was calculated. Individuals with this ratio at or above the sex-specific median were assigned a value of 1, and otherwise were assigned a value of 0. -The total score ranges from 0 (minimum adherence) to 9 (maximum adherence). *Ref: Trichopoulou A, Orfanos P, Norat T, Bueno-de-Mesquita B, Ocke MC, Peeters PH, van der Schouw YT, Boeing H, Hoffmann K, Boffetta P,* et al. *Modified Mediterranean diet and survival: EPIC-elderly prospective cohort study. BMJ 2005;330:991.* [68]	Ratjen et al., 2017 [53]
Dietary Approaches to Stop Hypertension (DASH)	-Consists of 8 components: vegetables, fruits, nuts and legumes, low-fat dairy products, whole grains, red and processed meat, sweets, and sodium -For fruits, vegetables, nuts and legumes, low-fat dairy products and whole grains, the lowest quintile of intake was given a score of 1, and highest quintile a score of 5. For sweets, red and processed meat, and sodium, for which a low intake is recommended, the scoring was reversed. -The score ranges from 8 (non-adherence) to 40 (perfect adherence). *Ref: Fung TT, Chiuve SE, McCullough ML, Rexrode KM, Logroscino G, Hu FB. Adherence to a DASH-style diet and risk of coronary heart disease and stroke in women. Archives of internal medicine. 2008; 168(7):713–720.10.1001/archinte.168.7.713* [69]	Izano et al., 2013 [44]; Fung et al., 2014 [52]; Guinter et al., 2018 [54]; Wang et al., 2020 [49]; Ergas et al., 2021 [50]
American Cancer Society nutrition score (ACS)	-The score was developed to evaluate the association of the ACS Nutrition and Physical Activity Guidelines for Cancer Prevention with cause-specific mortality. -Consists of 5 components: fruits, vegetables, whole grains, red and processed meat- It ranges from 0 to 9. It sums three key food-based recommendations (each contributing 0–3 points) with a score of 3 reflecting optimal adherence for each: “consume 5+ servings of a variety of fruits and vegetables”, “choose whole grains in preference to processed, refined grains”, and “limit consumption of red and processed meats” (quartiles of total red and processed meat, reverse-scored) *Ref: Kushi LH, Doyle C, McCullough M et al American Cancer Society Guidelines on nutrition and physical activity for cancer prevention: reducing the risk of cancer with healthy food choices and physical activity. CA Cancer J Clin 2012; 62:30–67* [70]	McCullough et al., 2016 [46]; Guinter et al., 2018 [54]; Al Ramadhani et al., 2020 [57]; Ergas et al., 2021 [50]
Empirical dietary inflammatory pattern score (EDIP)	-EDIP assesses the inflammatory potential of an individual’s diet. -It is a weighted sum of 18 food groups that are predictive of circulating inflammatory biomarkers, with higher (more positive) scores indicating more proinflammatory diets and lower (more negative) scores indicating anti-inflammatory diets. *Ref: Tabung FK, Smith-Warner SA, Chavarro JE, Wu K, Fuchs CS, Hu FB, Chan AT, Willett WC, Giovannucci EL. Development and Validation of an Empirical Dietary Inflammatory Index. J Nutr. 2016 Aug;146(8):1560-70.* [71]	Sasamoto et al., 2022 [58]
Australian Dietary Guideline Index (DGI)	-Consists of 15 components: vegetables and legumes, fruit, total cereals, meat and alternatives, total dairy, saturated fat, beverages, alcoholic beverages, sodium, and added sugars. Diet quality was incorporated using indicators relating to whole-grain cereals, lean meat, reduced/low fat dairy and dietary variety. *Ref: Sarah A. McNaughton, Kylie Ball, David Crawford, Gita D. Mishra, An Index of Diet and Eating Patterns Is a Valid Measure of Diet Quality in an Australian Population, The Journal of Nutrition, Volume 138, Issue 1, January 2008, Pages 86–93* [72]	Al Ramadhani et al., 2020 [57]
Chinese Food Pagoda-2007 (CHFP-2007), Chinese Food Pagoda-2016 (CHFP-2016)	-Consists of 10 components: salt, fats and oil, dairy products, beans, meat and poultry, fish, eggs, vegetables, fruits, grains -CHFP-2007 recommended intake amounts: salt (<6 g/d), fats and oils (<30 g/d), dairy products (>300 g/d), beans (>30 g/d), meat and poultry (<100 g/d), fish (>50 g/d), eggs (<50 g/d), vegetables (>400 g/d), fruits (>100 g/d), grains (>300 g/d) -CHFP-2016 recommended intake amounts: fats and oils (25–30 g), beans (25–35 g), meat and poultry (40–75 g), fish 40–75 g), eggs (40–50 g), vegetables (300–500 g), fruits (200–400 g), grains (250–400 g), the other components the same amounts as the CHFP-2007 -CHFP scores range from 0 to 45 points (lowest to highest adherence). *Ref: Wang SS, Lay S, Yu HN, Shen SR. Dietary Guidelines for Chinese Residents (2016): comments and comparisons. J Zhejiang Univ Sci B. 2016;17(9):649-656.* [73] *Yuan Y-Q, Li F, Wu H, Wang Y-C, Chen J-S, He G-S, Li S-G, Chen B. Evaluation of the Validity and Reliability of the Chinese Healthy Eating Index. Nutrients. 2018; 10(2):114.* [74]	Wang et al., 2020 [49]
Healthy Nordic Food Index (HNFI)	-Consists of 6 components typically consumed in Nordic countries: cabbage, root vegetables, rye bread, oatmeal, apples and pears, and fish and shellfish -A value of 1 is given for an intake at or above the sex-specific median of the sample and a value of 0 is given if the intake is below the sex-specific median for each item and each participant. -Score ranges between 0 and 6 (minimum to maximum adherence) *Ref: Olsen A, Egeberg R, Halkjaer J, Christensen J, Overvad K, Tjonneland A. Healthy aspects of the Nordic diet are related to lower total mortality. J Nutr 2011;141:639–44.* [75]	Ratjen et al., 2017 [53]

**Table 3 nutrients-15-03860-t003:** Assessment of Risk of bias based on the ROBINS-I tool.

*Study*	*Bias due to Confounding*	*Bias in Selection of Participants into the Study*	*Bias in Classification of Exposures*	*Bias due to Deviations from Intended Exposures*	*Bias due to Missing Data*	*Bias in Measurement of Outcomes*	*Bias in Selection of the Reported Result*	*Overall Bias*
** *Breast cancer* **								
2005; Kroenke [40]	Low	Low	Low	Low	Low	Low	Low	Low
2009; Kwan [41]	Low	Low	Low	Low	NI	Low	Low	Low
2011; Kim [42]	Low	Low	Low	Low	NI	Low	Low	Low
2011; George [43]	Moderate	Low	Low	Low	Low	Low	Low	Moderate
2013; Izano [44]	Low	Low	Low	Low	NI	Low	Low	Low
2014; George [45]	Low	Low	Low	Low	Low	Low	Low	Low
2016; McCullough [46]	Low	Low	Low	Low	Low	Low	Low	Low
2018; Sun [47]	Low	Low	Low	Low	Low	Low	Low	Low
2019; Karavasiloglou [48]	Low	Moderate	Low	Low	Low	Low	Low	Moderate
2020; Wang [49]	Moderate	Low	Low	Moderate	NI	Low	Low	Moderate
2021; Ergas [50]	Low	Low	Low	Low	Low	Low	Low	Low
** *Prostate cancer* **								
2014; Kenfield [55]	Low	Low	Low	Low	NI	Low	Low	Low
2015; Yang [56]	Low	Low	Moderate	Low	Low	Low	Low	Moderate
** *Colon cancer* **								
2007; Meyerhardt [51]	Low	Low	Moderate	Low	Low	Low	Low	Moderate
2014; Fung [52]	Low	Low	Low	Low	Low	Low	Low	Low
2017; Ratjen [53]	Low	Moderate	Low	Low	Low	Low	Low	Moderate
2018; Guinter [54]	Moderate	Low	Low	Low	NI	Low	Low	Moderate
** *Ovarian cancer* **								
2022; Sasamoto [58]	Moderate	Low	Low	Low	Low	Low	Low	Moderate
2020; Al Ramadhani [57]	Moderate	Moderate	Low	Low	NI	Low	Low	Moderate

Abbreviations: NI: No information.

## Data Availability

No new data were created. The data used in this study were found in published articles identified in the references. All results of meta-analyses are shown in tables and figures of the main text and Appendix A.

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
