# Peer review of "Post-Diagnosis Dietary Patterns among Cancer Survivors in Relation to All-Cause Mortality and Cancer-Specific Mortality: A Systematic Review and Meta-Analysis of Cohort Studies"

_nutrients, 2023, doi:10.3390/nu15173860_

Round 1
Reviewer 1 Report
This study is a systematic review and meta-analysis of cohort studies that synthesize the latest evidence regarding the association of a priori (diet quality indices) and a posteriori (data-driven) dietary patterns in relation to all-cause and cancer-specific mortality.
A total of 19 cohort studies including 38,846 adult cancer survivors, some assessing various dietary patterns, were included in the meta-analyses. Higher adherence to a priori DPs was associated with lower all-cause mortality by 22% (HR=0.78, 95% CI: 0.73-0.83, I2=22.6%) among all cancer survivors, by 22% (HR=0.78, 95% CI: 0.73-0.84, I2=0%) among breast cancer survivors and by 27% (HR=0.73, 95% CI: 0.62-0.86, I2=41.4%) among colorectal cancer survivors. Higher adherence to a “prudent/healthy” DP was associated with lower all-cause mortality (HR=0.79, 95% CI: 0.64-0.97 I2=49.3%), whereas higher adherence to a “western/unhealthy” dietary pattern with increased all-cause mortality (HR=1.48, 95% CI: 1.26-1.74, I2=0%) among all cancer survivors. Results for cancer-specific mortality were less clear. In conclusion, higher adherence to “healthy” dietary patterns, either a priori or a posteriori, was inversely associated with all-cause mortality among cancer survivors.
This is a well-documented and well-written paper, estimating pooled effect estimates of this meta-analysis, by random effects models to take into account the between-study heterogeneity.
Many papers reporting similar results have been published. Compared to previous meta-analyses, this meta-analysis includes solely observational studies, cohort studies in particular, and focuses exclusively on the post-diagnosis. Most randomized controlled trials conducted among cancer survivors investigating the role of diet in survivors’ prognosis so far have used indices of quality of life as primary outcomes and not survival. Carefully conducted observational studies, with low risk of bias, like most of the cohorts included in this meta-analysis, may provide reliable evidence on diet and health relationships.
This manuscript is ready for submission
Author Response
We thank the Reviewer for his review and we appreciate greatly his positive words and decision.
Reviewer 2 Report
A line in the Abstract conclusions giving the implications of your research would be welcome.
Introduction:
Lifestyle habits and modifications related to healthy diet and regular physical activity after cancer diagnosis are potentially important behaviors through which cancer survivors could protect and promote their well-being and longevity.
--
1. What is the main question addressed by the research?
To systematically review the evidence pertaining to dietary patterns and mortality outcomes among cancer survivors.
2. Do you consider the topic original or relevant in the field? Does it address a specific gap in the field?
Yes, this is an relevant topic and by focussing on current literature and looking only on post-diagnosis period adds to current knowledge.
3. What does it add to the subject area compared with other published material?
By focussing solely on post-diagnosis period and looking at cohort studies, it adds to the knowledge in this area.
4. What specific improvements should the authors consider regarding the methodology? What further controls should be considered?
The systematic review has been carried out using standard methodology and need not consider further controls.
5. Are the conclusions consistent with the evidence and arguments presented and do they address the main question posed?
Yes, conclusions are consistent with the evidence and arguments presented in the results and they address the main question posed.
6. Are the references appropriate?
References are appropriate.
7. Please include any additional comments on the tables and figures.
Tables and Figures are fine.
The paper would benefit from some minor English language editing.
Author Response
We appreciate greatly the Reviewer for his encouraging words and review.
1) As suggested, in order to show the implications of our research we have added one line in the abstract as follows:
"A “healthy” overall diet after cancer diagnosis could protect and promote longevity and well-being"
2) We have also tried to edit the whole text in relation to english language and triet to improved it as suggested by the Reviewer.